# Adherence to European Guidelines for Treatment and Management of Pancreatic Exocrine Insufficiency in Chronic Pancreatitis Patients

**DOI:** 10.3390/jcm10122737

**Published:** 2021-06-21

**Authors:** Mashroor Khan, Wiktor Rutkowski, Miroslav Vujasinovic, Johannes Matthias Löhr

**Affiliations:** 1Department of Medicine Huddinge, Karolinska Institutet, Huddinge, SE-141 86 Stockholm, Sweden; mashroor.khan@stud.ki.se (M.K.); miroslav.vujasinovic@ki.se (M.V.); 2Department of Upper Abdominal Diseases, Karolinska University Hospital, Huddinge, SE-141 86 Stockholm, Sweden; wiktor.rutkowski@sll.se; 3Department of Clinical Science, Innovation and Technology, Karolinska Institutet, Huddinge, SE-141 86 Stockholm, Sweden

**Keywords:** chronic pancreatitis, pancreatic exocrine insufficiency, chronic pancreatitis treatment guidelines, HaPanEU guidelines

## Abstract

European evidence-based guidelines for the treatment and management of chronic pancreatitis (CP) have been made available following the harmonizing diagnosis and treatment of CP across Europe (HaPanEU) initiative by the United European Gastroenterology (UEG). The aim of this study was to evaluate adherence to the guideline recommendations in the management of patients with pancreatic exocrine insufficiency (PEI) at Karolinska University Hospital in Stockholm. UEG guideline recommendations were evaluated and categorized into 55 different quality indicators (QIs). Data from a retrospective cohort of CP patients being treated at Karolinska University Hospital were evaluated with regard to overall adherence as well as adherence to specific QIs. A total number of 118 patients out of 956 patients diagnosed with CP were eligible for inclusion with mean overall adherence of 61.9% to the defined QIs. A significant difference in mean overall adherence was shown between patients diagnosed with CP prior to 1 January 2016 and following 1 January 2016 (59.3% and 67.7% respectively, *p* = 0.004), with linear regression analysis also demonstrating improvement correlating to date of diagnosis (*p* = 0.002). In conclusion, diagnosis and treatment of PEI improved after the HaPanEU guidelines became available and is continuously improving; however, there is room for further improvement.

## 1. Introduction

Chronic Pancreatitis (CP) is an irreversible, progressive and chronic inflammatory disease-causing life-altering disorders and disabilities in a significant proportion of affected individuals [1,2]. While epidemiologic data on CP patients are unreliable due to high variability in clinical presentation and difficulties in following long-term progression, salient studies have estimated an annual incidence of up to 13/100,000 people and a prevalence of 41/100,000 people [2,3]. In addition to reduced mean life expectancy with a median survival of 15–20 years following diagnosis [3], many CP patients suffer significantly reduced quality of life owing to the development of debilitating complications and sequalae, such as chronic pain, diabetes mellitus (DM), pancreatic cancer and pancreatic exocrine insufficiency (PEI), the latter being indicative of an advanced disease stage affecting more than 90% of the pancreatic gland [2,4]. Up to one-half of CP patients develop PEI with its clinical hallmarks of malnutrition, steatorrhea, weight loss and osteoporosis [5,6,7]. In order to address this multifaceted spectrum of disease presentation, progression, and complications, treatment of CP requires a multidisciplinary approach with the involvement of pharmacologic treatment, lifestyle modifications including dietary counseling for nutritional rehabilitation and possible surgical or endoscopic interventions [8,9]. Following years of increased understanding of, and improvement, in management of CP, an initiative from the United European of Gastroenterology (UEG) published evidence-based guidelines for diagnosis and therapy (HaPanEU) in 2017 [10]. The HaPanEU guidelines include 101 guideline recommendations covering 12 aspects of disease management. Previous studies have shown that PEI is often undertreated in CP patients [11], and a recent study evaluating guideline adherence in the Netherlands demonstrated suboptimal adherence to HaPanEU guidelines [12].

In this study, we evaluated adherence to, and the impact of promulgation of the HaPanEU guidelines in the treatment and management of CP patients at the pancreas outpatient clinic of the Department for Upper Abdominal Diseases at Karolinska University Hospital.

## 2. Materials and Methods

### 2.1. Study Design and Study Population

In this single-center retrospective cohort study, all patients with a definitive diagnosis of CP and PEI who were treated in the Pancreas Outpatient Clinic at the Department for Upper Abdominal Diseases of Karolinska University Hospital between 2016 and 2020 were eligible for inclusion. Diagnosis of chronic pancreatitis was determined by the M-ANNHEIM scoring system [13]. Patients having undergone pancreatic surgery prior to diagnosis, or other major abdominal surgery, for example colectomy, were excluded, as were patients who had been referred to other centers of specialty or primary care for further management, and patients with significant comorbidities.

### 2.2. Data Extraction and Parameters

In addition to retrieving data from previous existing data sets, local hospital electronic records, including journal entries, radiological evaluations, and laboratory analyses, were reviewed for further extraction of patient data. The existing data set included patient clinical characteristics at diagnosis such as age, gender, body mass index (BMI), disease etiology, smoking and alcohol status. In addition to these, the data set was expanded with extraction of data on specific symptomatology, therapeutic interventions and laboratory analysis, including stool frequency and characteristics, pancreatic enzyme replacement therapy (PERT), laboratory markers of malnutrition [14] including fat-soluble vitamin (A, D, E) and trace mineral levels (magnesium, calcium, iron), INR (as a surrogate for vitamin K that is not available in routine laboratory measures), albumin, cholesterol, cobalamin, and folate, as well as data on etiological screening and history.

### 2.3. Quality Indicators and Efficacy of Treatment

The HaPanEU guidelines were reviewed and classified into a list of 55 distinct quality indicators (QIs) (Appendix A). Collected data were evaluated with respect to defined QIs in order to assess how well patient care fulfilled evidence-based guidelines, including clinical and therapeutic recommendations. QIs were used and further clustered into six over-arching categories comprising: (i) patient screening for specific etiology of CP, (ii) screening for vitamin and trace mineral deficiencies, (iii) screening for pain, DM, osteoporosis and other CP complications, (iv) PERT supplementation, (v) supplementation of vitamin and mineral deficiencies as well as specific therapies targeted against known complications, including therapies for vitamin A, B, D, E and minerals such as calcium, folic acid, magnesium and iron, and (vi) treatment of other CP complications, such as pain, osteoporosis, treatment of concomitant diabetes mellitus as well as dietary counselling. Some quality indicators, such as screening for alcohol consumption were applicable for all patients, whereas other indicators, such as screening for genetic testing, were applicable only to a few patients.

The patients were divided into pre and postpromulgation groups (patients diagnosed with CP prior to or after 1 January 2016) to evaluate the impact of the guidelines on clinical practice. The treatment given to the prepromulgation group represented the baseline standard of care following an internal SOP, as the guidelines had not been published. The difference between the two groups signifies the impact the guidelines have had on treatment of patients with CP and especially PEI.

To determine the number of patients that had not been followed up, a patient lost to follow-up was defined as a patient either not having had a visit or phone call to a pancreatologist, or not having been screened for malnutritional values in the last 12 months.

To evaluate the efficacy of treatment of the HaPanEU guidelines, patients with abnormal lab values at their first screening who were treated according to the guidelines and had a second visit, were eligible for inclusion. Improved lab value at the second visit indicated efficacious treatment. All patients were seen by a group of four board-certified gastroenterologists. Patients seen by residents (follow-up visits only) were discussed with a gastroenterologist according to our internal SOP.

### 2.4. Statistical Analysis

Mean adherence to guidelines was defined by a percentage value of the given QIs for every patient added to a mean percentage value for the overall adherence to the QI’s. Separate percentage mean values were calculated in every given category for every patient. Percentage mean values were then compared between subgroups based on age group (<61 years or >61 years), sex category (female or male), year of diagnosis (prior to or after 1 January 2016) and etiology (toxic-metabolic, idiopathic/unknown, genetic, autoimmune, recurrent pancreatitis and obstructive causes respectively) [13].

Descriptive statistics were used for the categorical and quantitative variables describing cohort characteristics, including the estimation of mean values, range and standard deviations. Inferential statistics were calculated by testing for normality with the Shapiro-Wil test, followed by Student’s *t*-test for comparison between age group (<61 vs. >61 years) and sex (female vs. male). ANOVA tests were performed for analysis between etiological groups (Toxic-metabolic, Idiopathic/Unknown, Genetic, Autoimmune and Obstructive). Linear regression analysis was performed to determine the change in adherence ratio with regards to date of diagnosis. IBM SPSS Statistics 27 (IBM, Armonk, NY, USA) software was used for statistical analysis. The level of significance was defined as *p* < 0.05.

### 2.5. Ethical Considerations

The study was approved by the Regional Ethics Committee (Swedish: Regional Etikprövningsnämnden) in Stockholm, Dnr: 2020-02209. The requirement for individual informed patient consent was waived by the committee owing to the nature of a retrospective study, and that patients were not directly involved.

## 3. Results

### 3.1. Adherence to Guidelines and Treatment Efficacy

#### 3.1.1. Patient Inclusion and Demographics of the Patients 

Previous data sets were used for initial exclusion of noneligible patients. Of 956 CP patients being followed in the out-patient clinic, 246 were excluded due to no definitive CP diagnosis according to the M-ANNHEIM classification of CP. An additional 592 patients were excluded based on criteria specified above. In total, 118 patients fulfilled the criteria for inclusion and were included in the final cohort. A description of the inclusion process is detailed in Figure 1. A complete list of clinical characteristics of the cohort is given in Table 1. Of 118 patients, the majority were male (*n* = 72%), with a mean age at CP diagnosis of 59.7 years and a mean body mass index (BMI) of 23.2 kg/m². Almost one-half of patients (*n* = 54) were active consumers of alcohol at the time of diagnosis, with a slightly lower proportion of active smokers (*n* = 42). Similarly, about one-half of patients presented with known or newly discovered diagnosis of diabetes mellitus.

#### 3.1.2. Guideline Adherence

A summary of the overall mean adherence rates is presented in Table 2 with detailed data on adherence to specific QIs available in the Appendix A. The mean highest adherence was to PERT treatment being given according to guidelines, with an average adherence rate of 85.6%, followed by screening for malnutritional values (72.2%). The lowest adherence rate was in referral for dietary counselling (15.2%). As shown in Table 3, a significant difference was found in overall mean adherence between patients diagnosed prior to 1 January 2016 (the year the guidelines were completed) and patients diagnosed following 1 January 2016 (59.3% and 67.7% respectively, *p* = 0.004). A significant improvement in overall mean adherence rate in relation to date of CP diagnosis was demonstrated by the linear regression analysis shown in Figure 2 (*p* = 0.002). A total of 31 patients were lost to follow-up. The causes and date of their last visit are given in Table 4.

#### 3.1.3. Efficacy of Treatment

The efficacy of treatment is given in Table 5. Overall, few patients presented with abnormal lab values, and almost all patients who were treated according to the guidelines, improved their lab values, except for one patient who received vitamin D supplements and four patients receiving cobalamin supplements.

## 4. Discussion

To establish evidence-based guidelines is a laborious process. For chronic pancreatitis and pancreatic exocrine insufficiency, this did not happen in Europe prior to the UEG/HaPanEU guidelines [10]. It was the aim of this retrospective study to assess adherence to the guidelines and compare the quality of care before and after the availability of these guidelines in 2016 in a single-center retrospective cohort study. For the HaPanEU guidelines, a total of 101 questions covering 12 different topics, including diagnosis, therapy and treatment of the natural course of the disease, as well as common complications, were evaluated. In our study, these were categorized into a total of 55 different clinically relevant “quality indicators” (QIs), including clinical screening, laboratory analysis and documented therapeutic interventions. Our results demonstrate a mean adherence to guidelines of 61.9% for the complete cohort, and a significant difference between patients diagnosed with CP prior to guideline preparation in 2016 and patients diagnosed following 1 January 2016, with a mean adherence of 59.29% and 67.61% in these groups, respectively (*p* = 0.004). Linear regression analysis also showed a significant improvement over time (*p* = 0.002).

Our study suggests a rather significant impact on the management of patients diagnosed with CP before and after promulgation of the guidelines. There was also a steady increase in compliance to the guidelines. With some recommended interventions being more clinically relevant and important than others, adherence differed between the different QIs. Nevertheless, our study placed no weight on different QIs and, as such, the resulting mean adherence implies a substandard delivery of care. This is in line with a recent study [12] where a mean adherence rate of 53% was found for 93 patients with CP. With a similar method for the quantitative evaluation of guideline adherence, our study showed a superior adherence to QIs in a marginally higher number of patients. Our final cohort consisted of 118 patients, with the majority being male. This seemingly correlates with the higher proportion of Swedish males being exposed to known risk factors of CP, predominately smoking and alcohol [15]. The proportional etiologies among our patients are in line with subgroup expectations [16]. As such, our cohort is seemingly representative of CP patients in general.

Highest guideline adherence was to the recommendation of PERT treatment for improving stool symptoms and serum signs of malnutrition (85.6% mean adherence), followed by the recommendation for serum trace mineral deficiency screening (72.2% mean adherence). The benefits of PERT in improving serum nutritional parameters, gastrointestinal symptoms and quality of life have been demonstrated in several studies and have entered clinical routines at most major centers [17]. Screening for serum trace mineral deficiencies and other serum nutritional markers is facilitated locally by predefined groups of diagnostic tests in the local hospital electronic record system and ordered by the out-patient clinic nurses. In contrast, the lowest rates of recommendation adherence were demonstrated in factors reliant on the professional evaluation of the individual physician, such as proper documentation and screening of symptomatology and signs of complications. This implies the effectiveness of a team-integrative approach to treatment care, an approach that has not been extended to the inclusion of dietitians in patient care (15.8% adherence rate). Nevertheless, a multidisciplinary team-based approach has shown improved patient care in many other chronic conditions, as our study also implies [18]. Improvement of local routines and logistics could also potentially decrease the number of patients lost to follow-up. As such, it would seem that the development and continuation of local clinic routines are pivotal in ensuring proper patient care and follow-up.

Several biases were identified in the analysis of our study. Lack of documented symptomatology in the hospital electronic medical records meant a lack of data in several cases, although the relatively high number of included patients allowed for enough data points to yield significant results in several of our analytic models. Improving and facilitating data collection in itself, however, would allow for increased performance in screening for, and management of, CP complications. Developing standardized clinical instruments for patient screening and evaluation in clinical practice, such as patient questionnaires, could possibly improve assessment of patient symptomatology as well as serve as outcome indicators for evaluation of clinical practice, ultimately improving patient care [19]. Standardized PEI-specific evaluation tools have been suggested to improve identification, management and diagnosis of PEI, as well as assist in management of already diagnosed PEI [19].

Our results also indicate that patients diagnosed prior to 2016 displayed a lower mean adherence, which in part could be explained by a difference in clinicians’ approaches to first visits and follow-up visits. Several studies have demonstrated that clinicians perform a more thorough examination and history taking during the initial consultation compared with follow-up consultations [20]. Nevertheless, in other studies, those who receive PERT tended to demonstrate better compliance and receive a higher (more appropriate) dose of enzymes [21]. Improvement in physician adherence to guidelines does not necessarily correlate to improved treatment for the patients, because the patients could be noncompliant so that improved care does not occur. In our study, patients were met by a small team with virtually every physician meeting with every patient at one point in time and all patients were discussed at least once in a multidisciplinary team conference. As a result, we could not find any differences between individual physician’s adherence to the guidelines. However, compliance is difficult to assess. Recently, patient-reported outcome measures for PEI became available that can be used to address this issue [22].

Compared with similar studies, we were able to include a relatively high number of patients with confirmed chronic pancreatitis, meaning significant results could be obtained in our data analysis. In addition, extensive clinical evaluation and laboratory analysis was performed in most patients, meaning a vast number of variables and data could easily be obtained. Being a retrospective cohort study, intrinsic limitations to our study design meant a high reliance on previous documented data. As such, our outcome measure of guideline adherence as a relative proportion of fulfilled QIs could, in part, reflect the adequacy of clinicians’ documentation practices.

## 5. Conclusions

Following recently published guidelines on management of PEI in CP patients, the standard of care improved comparing standards before and after publication of the HaPanEU guidelines. Nevertheless, clinical adherence to HaPanEU guidelines remained relatively low (overall 61.9%, post-HaPanEU 67.7%), implying a suboptimal delivery of care to CP patients with PEI. However, adherence was seemingly higher compared with similar studies, possibly reflecting the proximity of the hospital to an academic research center with scientific proficiency. In addition, adherence improved over time, implying guideline awareness is increasing. Future patient care will most likely be more in line with guideline recommendations. Improvements in physician assessment and documentation, such as development of standardized instruments and questionnaires for patient screening and symptomatology, as well as better clinical routines and increased guideline awareness among treating physicians, could improve guideline adherence and provide better care to CP patients.

## Figures and Tables

**Figure 1 jcm-10-02737-f001:**
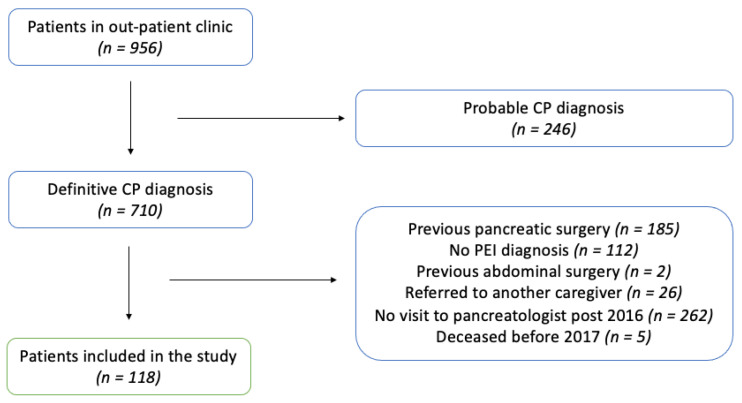
The process of patient selection.

**Figure 2 jcm-10-02737-f002:**
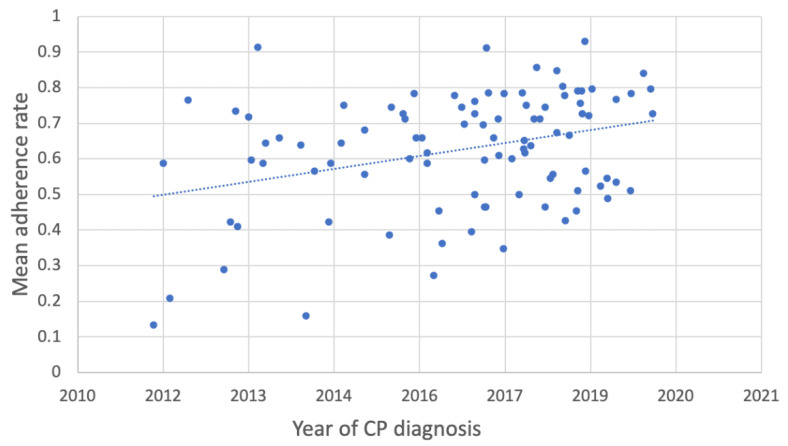
Linear regression of mean adherence rate in relation to date of diagnosis. X-axis represents the date of diagnosis, and the mean adherence rate is given on the y-axis (*p* = 0.002).

**Table 1 jcm-10-02737-t001:** Demographics and etiology of PEI patients according to the M-ANNHEIM classification of CP.

**Number of Patients (N)**		118
Demographics		
	Mean age (SD)	59.7 (12.8)
	Female sex (%)	33 (28.0)
	Male sex (%)	85 (72.0)
	Mean BMI (SD)	23.2 (3.5)
	Alcohol consumption at CP diagnosis (%)	54 (45.7)
	Previous alcohol consumption, before CP diagnosis (%)	12 (10.2)
	Smoker at CP diagnosis (%)	42 (35.6)
	Previous smokers (%)	44 (37.2)
	Smoking history; mean pack (SD)	29.9 (18.6)
	Never smoker	32 (27.1)
Etiology of CP (%)		N (%)
	Alcohol or nicotine consumption	68 (57.6)
	Hereditary factors	11 (9.3)
	Immunological factors	21 (17.8)
	Efferent duct factors	5 (4.3)
	Miscellaneous and metabolic factors	5 (4.3)
	Nutritional factors	1 (0.8)
	Unknown	7 (5.9)

PEI = pancreatic exocrine insufficiency; N = number of patients; SD = standard deviation; BMI = body mass index; CP = chronic pancreatitis.

**Table 2 jcm-10-02737-t002:** Overall mean adherence rate and mean adherence to the six general categories of quality indicators.

QI Category	Mean Adherence Rate % (SD)
Mean adherence rate for all QI	61.9
1. Screening for etiology of CP	61.0 (25.4)
2. Screening for abnormal malnutrition values	72.2 (10.0)
3. Screening for other PEI complications	46.4 (18.5)
4. PERT given according to guideline	85.6
5. Addition of supplements if abnormal nutritional values. Including PPI and PERT step up if insufficient PERT dosage.	57.1 (23.0)
6. Treatment of other PEI complications	61.8 (32.0)

QI = quality indicator; SD = standard deviation; CP = chronic pancreatitis; PEI = pancreatic exocrine insufficiency; PERT = pancreatic exocrine replacement therapy; PPI = proton pump inhibitor.

**Table 3 jcm-10-02737-t003:** Differences in total quality indicator mean adherence rate between cohort subgroups. Percentage in mean difference in case of etiology was calculated as the difference, and each was compared with the overall mean value.

Variable	Category	N (%)	Mean Guideline Adherence (%)	Mean Difference (%)	*p*-Value
Age	<61 years	55 (46.6)	60.4	2.8	0.34
	>61 years	63 (53.4)	63.2		
Sex	Female	33 (28.0)	60.0	2.6	0.21
	Male	85 (72.0)	62.6		
Etiology	Alcohol or nicotine consumption	68 (57.6)	60.9	−1.0	0.92
	Hereditary factors	11 (9.3)	61.0	−0.9	
	Immunological factors	21 (17.8)	62.4	0.5	
	Efferent duct factors	5 (4.3)	68.8	6.9	
	Miscellaneous and metabolic factors	5 (4.3)	62.1	0.2	
	Nutritional factors	1 (0.8)	72.7	10.8	
	Unknown	7 (5.9)	64.3	2.4	
Year of diagnosis	Before 2016	24 (20.3)	59.3	8.3	0.004
	After 2016	94 (79.7)	67.6		

N = Number of patients.

**Table 4 jcm-10-02737-t004:** Number of patients who were not followed up in the last 12 months, possible cause, and date of last visit.

**Patients Not Followed up in the Last 12 Months**	31
Cause	N
Refereed to/undergone imaging or interventional procedure and later not followed up	13
Visit/phone call in the last 12 months, but no new lab tests taken	8
Time booked, but patient did not show up to meeting	2
Patient unable to be reached on the phone	1
Unclear	6
Date of last visit	
Last 18 months	11
Last 24 months	9
Last 36 months	8
More than 36 months	3

N = number of patients.

**Table 5 jcm-10-02737-t005:** Efficacy of treatment.

Quality Indicator	Patients with Abnormal Values Treated According to Guidelines with a Second Follow-Up	Value Improved at Second Follow-Up (N)	Improvement in Lab Value (%)	*p*-Value
Vitamin and mineral				
Vitamin A	0	0	0	NA
Vitamin E	2	2	36	0.3
Vitamin D	6	5	196	0.02
Calcium	1	1	7.5	NA
Iron	4	4	70	0.02
Cobalamin	8	4	15.8	0.3
Folate	0	0	0	NA
Magnesium	2	2	6.7	0.29

N = number of patients.

## Data Availability

Our dataset contains sensible data which may contain private information about the patients treated at out clinic. The dataset can therefore not be made available to the public. However, the data used in our study can be provided upon request.

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
