# Peer review of "Adherence to European Guidelines for Treatment and Management of Pancreatic Exocrine Insufficiency in Chronic Pancreatitis Patients"

_jcm, 2021, doi:10.3390/jcm10122737_

Round 1

Reviewer 1 Report

The authors aimed to assesses as part of a MD thesis the compliance of patient care with European Guidelines for Treatment and Management of Pancreatic Exocrine Insufficiency in Chronic Pancreatis. The authors from a well established institution with a large specialised outpatient clinic observed that compliance with recently published guidelines on management of PEI in CP patients, is limited. This may suggest suboptimal delivery of care for patients with PEI. It is interesting to note that adherence appears to have improved over time. This is attributed to the implementation/use of standardised instruments and questionnaires that result in overall better and more standardised approaches. The manuscript reflects a solid clinical audit that may stimulate similar work elsewhere and helps to deliver high value care. Some questions remain

The authors' data  suggest that more recently diagnosed patients are more likely to receive therapy that is compliant with guidelines. Guidelines may have changed over time and adherence to guidelines may change over time. This should be clarified.

The quality of treatment and adherence to guidelines might be linked to differences in practice by clinicians. IS there an opportunity to explore this further?

Author Response

Thank you very much for your thoughtful comments and giving suggestions to improve our work. All authors have read your comments and made improvements in our text. We could address all issues raised.

Point 1: The authors' data suggest that more recently diagnosed patients are more likely to receive therapy that is compliant with guidelines. Guidelines may have changed over time and adherence to guidelines may change over time. This should be clarified.

Response 1: We agree.  This is the case indeed and is one of the main messages of the study. The manuscript was amended in two ways: In Methods section (lines 107-112 and 120-122) and in the Discussion section (lines 298-300).

Point 2: The quality of treatment and adherence to guidelines might be linked to differences in practice by clinicians. IS there an opportunity to explore this further?

Response 2: We found no difference, and we clarified this both in the Methods (see above) and discussion (lines 258-261 and 329-333).

Reviewer 2 Report

Suggestions to improve the manuscript:

  1. There is considerable evidence that the timeline for practice change in response to new evidence is very long.  Perhaps it was the authors intent but the presentation leads to a set of conclusions that are primarily negative and could be more positive.  An alternative take might be to make this a analysis of the impact of new guidelines (in a relatively short time period) on care.  Instead of suggesting that, in the first group, the study is measuring compliance with guidelines that have not yet been formulated, the authors might suggest they are measuring the impact of a set of guidelines on care before and after their promulgation.  In the prepromulgation group you are not measuring compliance but rather the baseline care standards.  The results then suggest a relatively rapid improvement in care as the result of new guidelines, acknowledging that there is still a ways to go.
  2. The authors conclude that their results imply "a suboptimal delivery of care to patients with PEI."  This cohort of patients is one in whom compliance with recommendations may be the major issue.   Do the authors have any data on the difference between recommended care, which one might expect would be more in line with the guidelines, versus patient compliance with recommendations?  If not, this should be clearly stated in the limitations.

Author Response

Thank you very much for your thoughtful comments and giving suggestions to improve our work. All authors have read your comments and made improvements in our text. We could address all issues raised.

  1. There is considerable evidence that the timeline for practice change in response to new evidence is very long. Perhaps it was the authors intent but the presentation leads to a set of conclusions that are primarily negative and could be more positive. An alternative take might be to make this a analysis of the impact of new guidelines (in a relatively short time period) on care. Instead of suggesting that, in the first group, the study is measuring compliance with guidelines that have not yet been formulated, the authors might suggest they are measuring the impact of a set of guidelines on care before and after their promulgation. In the prepromulgation group you are not measuring compliance but rather the baseline care standards. The results then suggest a relatively rapid improvement in care as the result of new guidelines, acknowledging that there is still a ways to go.

Response 1: Thank you for this very good suggestion. We amended the text accordingly. Specifically:

  • Abstract; we have changed the conclusion part to say that the adherence has improved significantly over time (lines 24-26).
  • Introduction; mentioned that our aim was to evaluate the impact of these guidelines in our clinical practice (line 63).
  • In Methods; specified a pre-promulgation and post-promulgation groups and that we are measuring the difference between the groups (lines 107-112).
  • We have put more emphasis on the difference between the two groups in the Discussion (lines 244-248; lines 259-262).
  • In the Conclusion, we have put more emphasis on the rapid increase in adherence (lines 344-347).
  1. The authors conclude that their results imply "a suboptimal delivery of care to patients with PEI." This cohort of patients is one in whom compliance with recommendations may be the major issue. Do the authors have any data on the difference between recommended care, which one might expect would be more in line with the guidelines, versus patient compliance with recommendations? If not, this should be clearly stated in the limitations.

Response 2: We agree that compliance is a major issue in this cohort of patients. Unfortunately, we were not evaluating the patient compliance to the guidelines in this project, and investigate how well the physician’s treatment and management of the patients adhered to the guidelines.

  • We have now mentioned that improved physician adherence does not necessarily lead to improved care for the patients as a limitation to our study in discussion (line 330-337).
  • We have rewritten parts of our conclusion and removed the part “suboptimal delivery of care to patients with PEI” and rewritten the conclusion part to emphasize the rapid increase as mentioned earlier (line 347-350).

Round 2

Reviewer 2 Report

No further suggestions of authors